# Repeated Palaeofloods of 8.2–6.4 ka and Coeval Rise of Neonatal Culture in the Upper Yangtze River, China

Zhongxuan Li [1],* and Wenhao Li [2]

1   School of Urban and Environmental Sciences, Xuchang University, Xuchang 461000, China
2   School of Energy Science and Engineering, Nanjing Tech University, Nanjing 211816, China
*   Correspondence: aysylzx@163.com

**Abstract:** Flood events have long been very frequent along the Yangtze River in Chongqing, China. A complete sedimentary sequence of alluvia, found in the Yuxi profile (YXP) was applied to explore features of the palaeoflood layers that maintained records related to the contexts of flooding hydroclimate. The AMS[14]C dating results dependent on animal bones from the YXP validate that the chronology of the palaeoflood layers was dated, between ca. 8200 and 6400 a BP, and multiple cultural layers were intercut among these palaeoflood layers. By means of particle size and end-member analyses for the palaeoflood sediments, the fractions of fine silt and clay in deposits account for a high proportion of the flood sediments, suggesting that the overbank flood was the main power in building the palaeoflood layers. Due to the climatic episodes defined by pollen assemblages, the thickness of the flood layers is positively correlated with soil erosion because of different hydrothermal conditions. The wavelet spectra of the mean particle-size series also suggest that there may be two major palaeoflooding cycles of ~700 and ~30 years. Despite the sustained palaeoflooding, the Yuxi Culture grew from small to big, and was never broken off, in terms of the findings of artificial remains in the YXP.

**Keywords:** palaeoflood; sedimentary layers; climatic change; Yangtze River; Yuxi Culture of Chongqing





## 1. Introduction

Close linkages between climatic cooling and sustained palaeoflooding since entry into the Holocene (ca. 11.7 ka BP; BP is short for before the present, i.e., before AD 1954 and the same below) have long been fully corroborated [1–4]. As bedded flood layers are often left after flooding, we are able to unravel accurate features in hydrodynamics, discharge-level, and periodicities of repeated palaeofloods [5–7]. To better understand human responses to a variety of palaeofloods, archaeological sites have been, over the past decades, greatly attached to explaining the interplay between Earth and Man [8–10]. At a typical site, fluvial deposits could, in general, reserve relatively accurate tracks of climate change, as well as human activity [11], which naturally provide us more faithful means to fully describe what the scenarios of climate and society were like.

A general agreement has been reached that declines that Neolithic cultures across the Earth were commonly connected with climatic deteriorations [12–19]. A growing number of standpoints [20–22] for humans on the adaption to adverse social environments is, however, widely accepted in contrast to environmental determinism. Previous research on palaeofloods was reported now and then, but still left two major issues unaddressed. First, many documents attached importance to the macro-regional research into palaeoenvironmental indicators recorded either by alpine peat [23–26] or by karst stalagmite [27–29], which cannot fully expound the endemic variation of hydrothermal conditions. Second, the existing studies [30–35] have long been focused on the negative effects of extreme climate events, ignoring the value of human response to environmental strikes. Contrary to previous studies [36–38] that were inclined towards the mid-reach rivers with dense sites

of human–environment interactions, this paper lays emphasis on the evidence compiled from the upper-reach of the Yangtze River.

Recent bicentennial flood record documents indicate that the Chongqing region was in a period of frequent floods in the nineteenth century [39], which was exactly at the same time as the global Little Ice Age. Yao et al. [40] suggest that the multi-period flood events in that century are highly correlated with the increased accumulation of the Dassop glacier in the Himalayas, when the Indian monsoon was bringing abundant rainfall. According to the records of the Cuntan-gauged station [39], four super floods (1840, 1870, 1905, 1981) with the highest peak discharge exceeding 80,000 $m^3$/s in the Yangtze River section of Chongqing since 1800 were experienced. The maximum peak discharge in 1870 reached 96,000 $m^3$/s, and the water level rose up to 192.2 m, causing disasters across 20 counties, reflecting that regional flooding is a long-standing issue.

Allowing for the alluvia-based profile at an archaeological site, our hypotheses are that: (1) Sedimentological analyses of deposits can rebuild the occurrence in which palaeofloods were processing, as well as a high correlation between pollen assemblages and local vegetation; (2) There is a significant coupling between palaeoflood invasions and human retreats; (3) The Yuxi residents had merely primitive resources available and limited development strategies. Hence, this paper mainly comprises three sections: (1) Building an R-program-based age frame of the stratigraphic sequence and making a detailed description of flood layers, which allow us to deeply investigate the sedimentary characteristics; (2) Exploring the congruent relationship between the flood layers and climatic variation, which is supported by particle size, magnetic susceptibility, ratios of Rb/Sr and Si/Al, and composition of pollen assemblages; (3) Unveiling the subsistence regime of an ancient society to better understand how societal resilience was maintained when confronting sustained flooding.

This study resultantly aims to: (1) Investigate the sedimentological features of the palaeoflood layers based on the particle-size distribution of the flood sediments; (2) Discuss the identified climatic phases using environmental proxies, e.g., the percentage of pollen assemblages, the ratios Rb/Sr and Si/Al; (3) Explore the likely cycles of the palaeoflooding occurrence by means of wavelet spectral analysis of the sedimentary sequence; (4) Explain interactions between humans and floods according to unearthed artifacts.

## 2. Materials and Methods

Prior to collecting the soil samples, a thorough field work of identifying the layers in the Yuxi profile (YXP) with a total depth of 665 cm was initially conducted by means of archaeology and geology. The layers were cut and collected using six head-to-tail one-meter-long steel sampling boxes, and the retrieved stratigraphic column immediately sealed in plastic wrap was taken back to the Institute of Regional Environmental Evolution, Nanjing University. For different analytic aims, the soil column was segmented in unequal space. A total of 497 samples were applied for measuring magnetic susceptibility, 260 samples for the examination of particle size, and three groups comprising 48 parallel samples for identifying pollen species, testing the contents of macro-elements, respectively. In addition, four samples of flood layer formed in 2004 and 2008 were collected to act as the comparative reference.

Air-dried and treated by diluted HCl (10%) and HF (10%) to remove $CaCO_3$ and organic materials coated on soil particles, the samples were used for particle-size measurement with a Malvern Mastersizer 2000 laser analyzer. The magnetic susceptibility of a soil sample with a mass of ten grams was measured using an AGICO KLY-3 magnetic susceptibility meter (875 HZ, 300 A/m) at the Institute of Regional Environmental Evolution, Nanjing University. To determine the concentrations of major elements, a dried sieved sample with 0.5 g under 100 mesh was in turn treated using three acids, i.e., HCl (10%), HF (10%) and $HClO_4$ (5%); after being heated and steamed dry, the solution was extracted by the method of diluted $HNO_3$ (5%), and then element concentrations were determined with an inductively coupled plasma atomic emission spectrometer (Perkin Elmer 3300RL),

whose effective dose rate was determined with the elemental concentrations by using the revised dose-rate conversion factors [41]. After a process of powder-compressing, the contents of Rb and Sr were measured with an ARL-9800 X-Ray fluorescence spectrometer at the Center of Modern Analysis, Nanjing University. A chronological framework for the YXP was established by pedostratigraphic correlation with the dated bone samples from Layers 7, 9, 11, 15, 19, 21, and 27 by means of AMS[14]C radiocarbon dating at the Institute of Earth Environment, Chinese Academy of Science. The final age data were calibrated in the software CALIB 5.0.1 [42] conjunct with the INTCAL 09 dataset [43]. A specific time-scale of the stratigraphical sequence for the YXP was conducted using the radiocarbon AMS[14]C data through age–depth modelling in R-language-based *Bacon* [44], using a smoothed spline regression. End-member estimation and wavelet spectral analysis for the palaeoflood deposits, additionally, were processed in Matlab-based calculating programs. To identify the pollen taxa in the soil samples, each sample (50 g in weight) was treated through a standard procedure of NaOH (5%), HCl (10%), and HF (40%) to remove impurities. After removing water with a centrifuge (for 5 min at 2500 rpm), the treated samples were reserved in glycerin for identification under a microscope. The tech route flowchart is seen below in Figure 1.

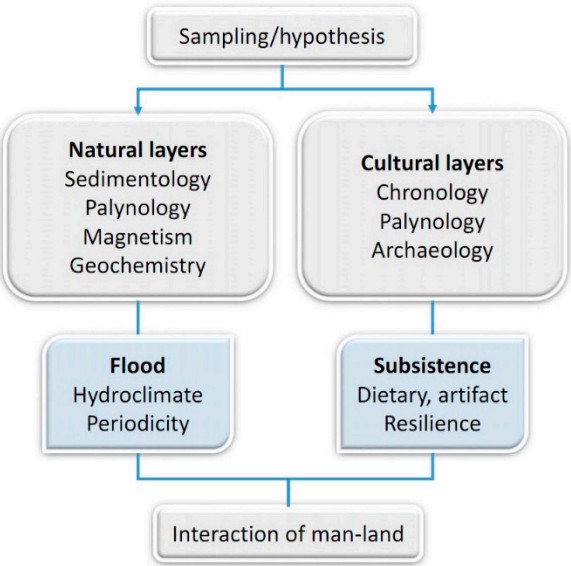

**Figure 1.** Flowchart of the tech route for this study.

## 3. Geographical Settings

The Yuxi archaeological site (Figure 2), covers over 80,000 m$^2$, its elevation is between 153–175 m above the Wusong horizontal zero and it is situated on the second terrace of the east bank of the Yangtze River, Fengdu County of Southwestern China. The site's central geographical coordinates are 30°02′14″ N, 107°51′38″ E. This area is characterized by a subtropical monsoon climate with distinct seasons in a year, mean annual temperature ranging 17.2–18.5 °C, and mean annual precipitation between 990 and1120 mm. Deeply influenced by the warm currents from Indian and Pacific Oceans, it is rainy in the summer and autumn; due to the confluence of the rivers, the Yuxi site is often subject to extreme flooding. Additionally, the confined terrain of the narrow river valley of the Yangtze River and a sustained rainfall always prompt the flood water level to exceed 190 m, e.g., 191.5 m gauged at the Cuntan station in 2020. As above, the Yuxi archaeological site has long been subjected to frequent flooding.

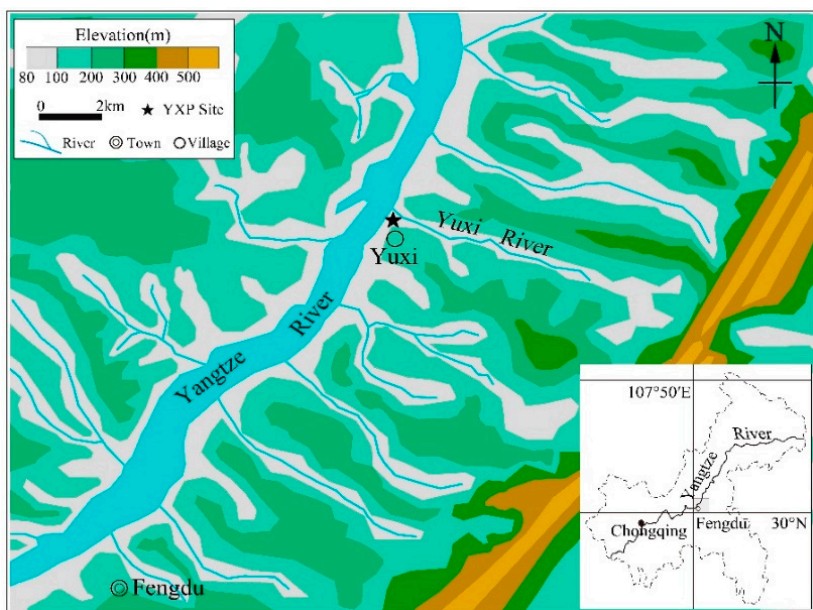

**Figure 2.** Map showing the geographical location of the Yuxi archaeological site.

The regional vegetation is mid-subtropical complex evergreen forests with acid yellow soil. Because of the long-term interruption from rural activities, there are two common forests, i.e., evergreen broad-leaved and conifer broad-mixed. The former has the dominant species of evergreen chinquapins (*Castanopsis fargesii*), walnuts (*Juglans regia*) and horn beams (*Carpinus*). The latter, as a secondary community, has widely dispersed species, predominantly pines (*Pinus massoniana*) and bamboos (*Phyllostachys glauca*). Herbal vegetation consists of compositae, artemisia, and nettle. In addition, the zonal ferns include *Pteridaceae*, *Polypodiaceae*, and *Lycopodium*. All the above generally constitute the zonal landscape of plants.

## 4. Results

### 4.1. Stratigraphy of the YXP Layers

Together with archaeological artifact identification, sedimentological criteria (e.g., sedimentary properties in color, texture, and structure) partition the Yuxi stratigraphic layers (Figures 3 and 4a) into 31 sub-layers. Layers 4 to 30 are a complete sedimentary sequence in chronological continuity (~8.2–4.8 ka BP). Human ruins, including potteries, burnt earth, and animal bones were sporadically found in the cultural layers, mainly made up of talus deposits. Archaeologically, the stratigraphic sequence below Layer 8 is named as the lowest stage of the Yuxi Culture (~7.8–6.3 ka BP, Figure 4a) and the intermittently fluvial deposits consist of clayey silt, mealy sand, and clay. The thickness of the eleven palaeoflood layers ranges from 5 to31 cm. Regarding the contiguity of the layers, those in large inclination with clear beds and in good contact with mutuality are readily distinguishable to elucidate their primary sequential properties. Because our focus is mainly on the natural palaeoflood layers, this paper will concentrate on the lower stage of the Yuxi Culture, where the palaeoflood layers are especially centered on. A brief description for Layers 8–31 is exhibited in Figures 3 and 4a.

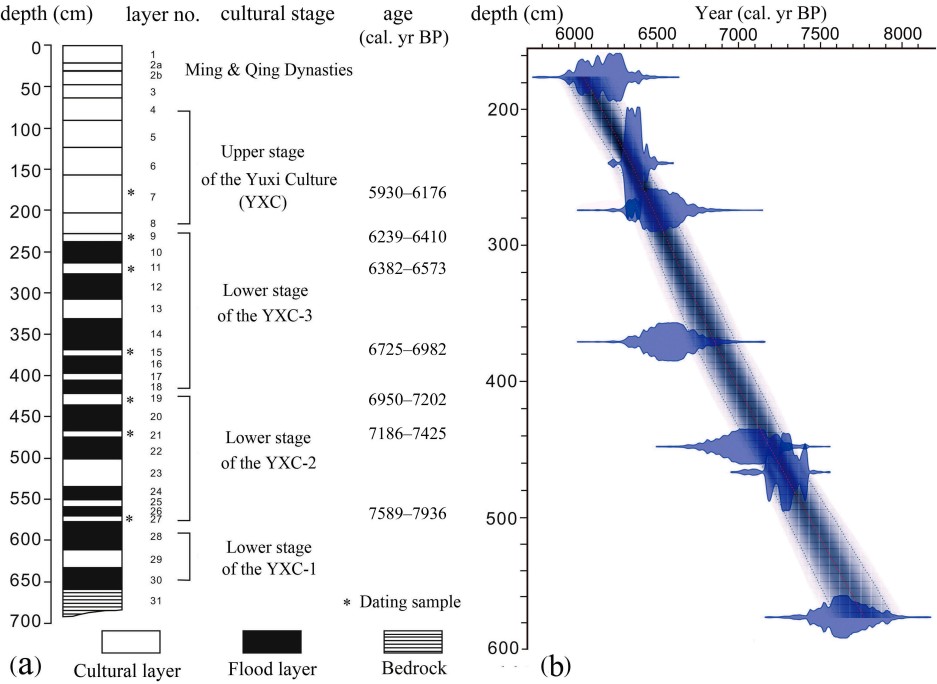

| Layer No. | Thickness (cm) | Lithofacies | Cultural age | Brief description of sediments at Yuxi profile | Simulated age (ka BP) |
|---|---|---|---|---|---|
| 8 | | | | Turquoise hard soil containing many ginger rocks | 6.2 |
| 10 | 81 | | Late Stage | **cultural layer**: grey soft soil containing fish bone residues and large animal skeletal remains | 6.3 |
| 12 | | | | **natural layer**: Light yellow sandy clay containing a small amount of burnt soil particles | 6.6 |
| 13 | 92 | | Middle Stage | **cultural layer**: grayish silty clay containing a few of bone shards, burnt earth particles | 6.7 |
| 14 | | | | **natural layer**: taupe clayey sand containing amounts of charcoals | |
| 16 | | | | | 6.9 |
| | | | | | 7.0 |
| 20 | | | | | |
| 22 | | Early Stage (Lower Yuxi Culture) | | **cultural layer**: greyish silty clay containing large amounts of ashes, bone shards, artifacts; mahogany silt containing a few charcoals, antler-bones | |
| 23 | 265 | | | | 7.6 |
| | | | | **natural layer**: cyanic clayey silt without other inclusions; yellowish brown silt with sandy gravels containing a bed of charcoals; light yellowish clay containing a few charcoals | |
| 28 | | | | | |
| 29 | | | | | |
| 30 | | | | | 8.2 |
| | | bedrock | | △ pottery  I animal bone  ------ charcoal  ┊ burnt particle<br>◇ pebble  ≪ fish bone  ○ stone ware | |

**Figure 3.** Generalized lithostratigraphic diagram of the sediments at the Yuxi archaeological profile showing a stratigraphic contact relationship and a brief description of lithofacies.

**Figure 4.** Lithological diagram of the Yuxi archaeological profile. Panel (**a**) shows the layer sequence and cultural stages; panel (**b**) shows the fitting relationship between sediment accumulation rate and calibrated years based on age-depth modelling by Blaauw et al. [44].

*4.2. Chronology*

Temporal control of the sedimentological strata in the YXP was established based on seven AMS[14]C dating data determined from bone fragments listed in Table 1. The sample from Layer 9, bordered by the topmost flood Layer 10, was in the AMS[14]C radiocarbon age dated 5567 ± 40 yr BP at the depth of 241 cm (calibrated 6350–6306 a BP) and the sample from Layer 27 dated 6795 ± 123 yr BP at 582 cm (calibrated 7749–7565 a BP). As there is an absence of dating materials in these natural layers, it is, hence, required to use the age–depth modelling performed in *Bacon* to simulate a time reference for the sedimentological sequence. According to the top and bottom limits of the modelling years of the palaeoflood layers, they were determined to ca. 6400 and 8200 a BP, achieved through the sediment accumulation modelling (Figure 4b) performed in *Bacon* [44]. In addition, the YXP has three evident features in the stratigraphical sequence: (1) The palaeoflood layers alternate with the cultural layers with bone residues for dating; (2) The continuity of the study layers (238–663 cm) yields a correlation between year and depth; (3) The interval of the palaeoflood layers falls exactly in 6400–8200 a BP, in line with that of the palaeofloods that occurred at the Chengbeixi Site and the Guanzhou Site, downstream of the Yuxi Site [45,46].

**Table 1.** AMS[14]C ages of the dated cultural layers at the YXP.

| Layer No. | Lab No. | Material | AMS[14]C a BP * | Calibrated (Agecal a BP) | |
| --- | --- | --- | --- | --- | --- |
| | | | | 1σ | 2σ |
| 7 | XA57 | Bone | 5398 ± 84 | 6287–6169 | 6312–5986 |
| 9 | XA58 | Bone | 5567 ± 40 | 6350–6306 | 6410–6288 |
| 11 | XA59 | Bone | 5709 ± 90 | 6570–6407 | 6669–6308 |
| 15 | XA60 | Bone | 5773 ± 100 | 6667–6467 | 6758–6394 |
| 19 | XA61 | Bone | 6168 ± 99 | 7163–6939 | 7268–6846 |
| 21 | XA62 | Bone | 6365 ± 55 | 7325–7249 | 7422–7207 |
| 27 | XA63 | Bone | 6795 ± 123 | 7749–7565 | 7844–7431 |

* The [14]C ages above have been regulated by CALIB 5.0.1 [42].

*4.3. Palaeoclimatic Phases*

Some chemical elements have differentially fractionated features in supergene geochemical processes, for example, rubidium, strontium, silicium, and aluminum; their ratios, Rb/Sr [47] and Si/Al [48], are often employed to report chemical weathering of the top soil related to eluviation resulted from hydroclimatic variation. We present here the variation of Rb/Sr and Si/Al ratios along the sedimentological sequence, together with curves of magnetic susceptibility subdividing the sedimentological interval (~6400–8200 a BP) into three hydrothermal phases (Figure 5): (i) The warm-wet phase (450–671 cm), characterized by the intensive weathering evidenced by the high value of mean magnetic susceptibility at 70.9 SI ($10^{-8}$ $m^3$ $kg^{-1}$) and the highest values of Rb/Sr ratios and Si/Al ratios(470–671 cm); (ii) The temperate dry phase (348–450 cm), characterized by a much weaker chemical weathering of the hydroclimatic regime, with a significant decrease in magnetic susceptibility values and an undulant reduction in Rb/Sr and Si/Al ratios; (iii) The warm-dry phase (220–348 cm), characterized by a revival of the chemical weathering process as the magnetic susceptibility values in trough compared to the high-value interval in Rb/Sr and Si/Al ratios.

Pollen assemblages extracted from terrigenous strata can serve as an effective indicator to the climatic regime of a catchment [49,50]. Due to uniformitarianism, a territory where trees and ferns are thriving is assumed to be humid, whilst an increased herbal percentage indicates that a region is surely drying. The pollen composition in Figure 5 shows the average percentage of the trees in the warm-dry phase (ca. 6300–6800a BP), which is 43%, for herbs 37.9%, and for ferns 18.6%. During the temperate dry phase (ca. 6800–7200 BP), the pollen components are 62.9% trees, 24.6% herbs, and 12.5% ferns, suggesting an increase in tree percentage and declines in herbs and ferns. The warm-wet phase (ca. 7300–8200 a BP), consists of 49.1% trees, 31.5% herbs, and 19.4% ferns, reflecting the Yuxi area was warm-wet

in the early Yuxi Culture. Such a hydroclimate variation in palynology, taken overall, is substantially consistent with what is seen by the changes in magnetic susceptibility, Rb/Sr, and Si/Al ratios.

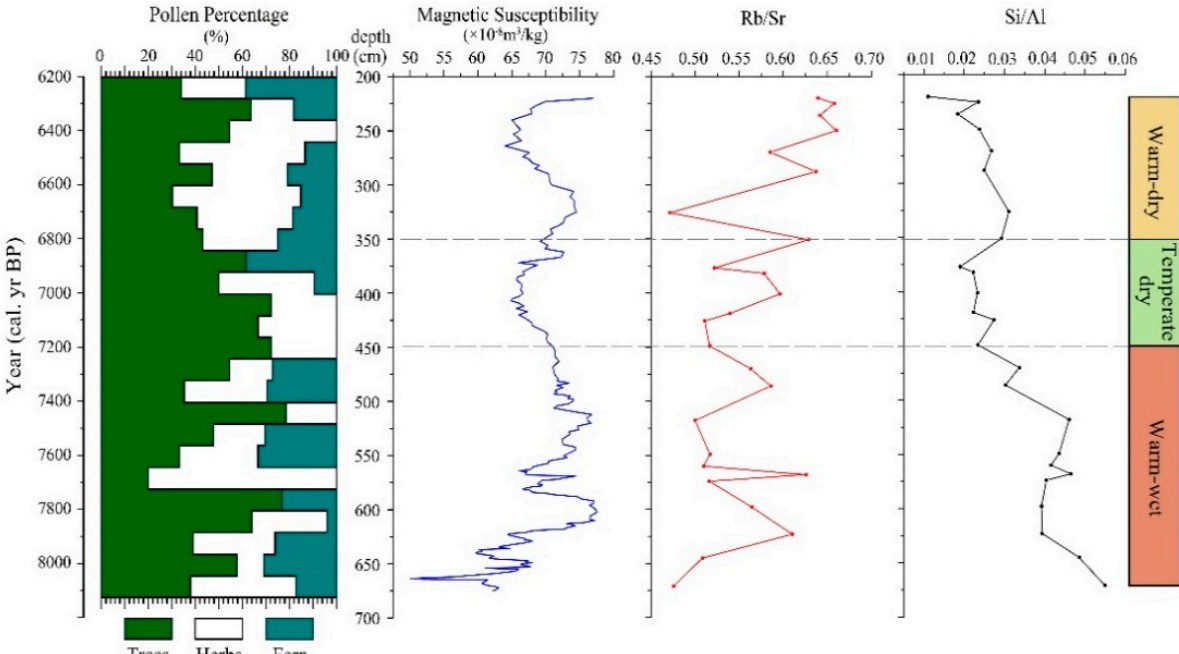

**Figure 5.** Three climatic phases divided based on the proxies of pollen percentages, magnetic susceptibility, Rb/Sr and Si/Al ratios, indicating the different hydrothermal contexts.

*4.4. Identification of the Palaeoflood Layers*

The most prominent features for the YXP, exhibited in Figures 3 and 4a, are the natural layers (even-numbered below layer 9), which are relatively uniform in soil components, but the cultural layers (odd-numbered below layer 9) contain different-sized sediment mixture with wormholes, crevices, burnt earth particles, and pottery shards; two types of layers were alternately constructed. The natural layers, consisting of yellowish cinnamon coarse silt (17.1–34.2 μm) at the bottom and brownish sandy clay at the top, are greater in thickness ranging from 10 to 35 cm than the cultural layers (10–25 cm). In terms of the sediment features of the layers in earth color, structure, texture, compactness, and particle-size composition, there are clear boundaries between natural and cultural layers. Together with the sporadic mud cracks discerned on the top of the natural layers [51], these distinguishable natural layers in the YXP are reliably formed by a vertical accretion of palaeoflood deposits.

Poorly sorted palaeoflood deposits in the YXP are suggested by sorting coefficients (Sc > 1) in Table 2 associated with the very close values in kurtosis and skewness, which result in unimodal distribution and positive skew, meaning a roughly consistent sedimentary setting. Meanwhile, the mean particle size and skewness of sediments of Layers 14, 16, 18, 20, 22, and 24 are strikingly larger than their neighboring layers, suggesting that these six layers, when formed, underwent drastic climatic changes. With the aforesaid climate phases, the six layers fall exactly in the 270–522 cm interval that corresponds to the temperate-dry phase, meaning a heavy flood time, probably linked to the reduction in vegetation cover and the enhancement of water erosion. Despite the modern flood deposits (e.g., the flood in 2004), the mean particle size (Table 2) is coarser than the palaeoflood ones; their major parameters in particle-size distribution, on the whole, substantially fall in identical intervals.

**Table 2.** Particle-size parameters of the natural deposits compared with the 2004 flood deposits in the YXP.

| Layer No. | Mean Grain-Size/Mz (μm) | Standard Deviation/σ | Skewness/Sk | Sorting Coefficient/Sc | Kurtosis/Ku |
|---|---|---|---|---|---|
| 10 | 19.64 | 1.33 | 0.18 | 1.95 | 0.99 |
| 12 | 17.82 | 1.41 | 0.20 | 1.99 | 0.94 |
| 14 | 34.20 | 1.09 | 0.28 | 1.78 | 1.15 |
| 16 | 23.85 | 1.18 | 0.28 | 1.81 | 1.06 |
| 18 | 21.94 | 1.34 | 0.22 | 1.94 | 0.97 |
| 20 | 29.56 | 1.36 | 0.22 | 2.00 | 1.00 |
| 22 | 25.03 | 1.28 | 0.24 | 1.92 | 1.03 |
| 24 | 29.16 | 1.20 | 0.22 | 1.76 | 1.00 |
| 26 | 18.20 | 1.23 | 0.08 | 1.77 | 0.97 |
| 28 | 17.10 | 1.33 | 0.05 | 1.95 | 1.00 |
| 30 | 24.86 | 1.35 | 0.16 | 1.98 | 1.01 |
| 31 | 25.03 | 1.33 | 0.17 | 1.98 | 1.02 |
| 2004 (1) | 53.66 | 1.35 | 0.28 | 1.59 | 1.25 |
| 2004 (2) | 51.83 | 1.42 | 0.25 | 1.33 | 1.13 |
| 2004 (3) | 37.68 | 1.36 | 0.34 | 1.66 | 1.19 |

*4.5. Particle-Size Distribution of Palaeoflood Sediments*

The particle size of deposits can generally play a major role in deciphering the sedimentary environment [52,53]. As described in Figure 6a,b, the particle size of the palaeoflood deposits in the YXP mainly ranges between 2 and 63 μm. Exceptionally, the modes of the volume percentage of the deposits in layers 10, 12, 14, 16 and 22 fall in the range of 20–80 μm, partly overlapping the sand-grained interval, and the cumulative probability shows that the sand population contributes over 23.5% to the palaeoflood sediments, while the clayey fraction acts as a minor role. Accounting for the largest population in the deposits, the fine silt and clay populations (the suspended loads) evidently take control of the total palaeoflood formation, and the sand population merely has the saltation role. The traction loads in the palaeoflood layers are absent.

End-member (EM) analysis developed by Paterson and Helslop [54] was employed to describe the dynamics of the sediment delivering. On the premise of satisfying weaker correlations ($R^2$ less than 0.5) between each member and the curve of angular deviation with a flat slope (Figure 6c), the parameterized EM analysis was conducted with the AnalySize Software Package [54]. Figure 6d illustrates the five separated EMs for the palaeoflood deposits in the YXP. EM1 and EM2 in Figure 6d peak at 2 μm and 18μm, respectively, being the clayey silt fraction, reflecting standing-water flood deposits (Table 3). The coarse sand fraction is suggested by EM5, reflecting a fast-moving flood context. Two transitional components, EM3 and EM4 that belong to the fraction of silt and fine sand, give the context of being transported by a slow-moving water flood that was probably dammed by the bank or a tributary nearby. It is, therefore, suggested that the three types of delivery powers for the palaeoflood deposits are the standing water, slow-moving water, and fast-moving water of overbank floods, which represent different sedimentary environments when the palaeoflood layers were formed.

**Table 3.** Particle composition of end-members separated from the palaeoflood deposits.

| End Member | Clay/% | Silt (%) | Fine Sand (%) | Coarse Sand (%) | Sediment Carriers |
|---|---|---|---|---|---|
| EM1 | 64.7 | 33.3 | 1.8 | 0.2 | Standing water |
| EM2 | 9.6 | 82.0 | 8.3 | 0.1 | Standing water |
| EM3 | 0.3 | 65.6 | 34.1 | 0 | Slow-moving water |
| EM4 | 0 | 31.9 | 66.6 | 1.6 | Slow-moving water |
| EM5 | 0 | 0 | 25.9 | 74.1 | Fast-moving flow |

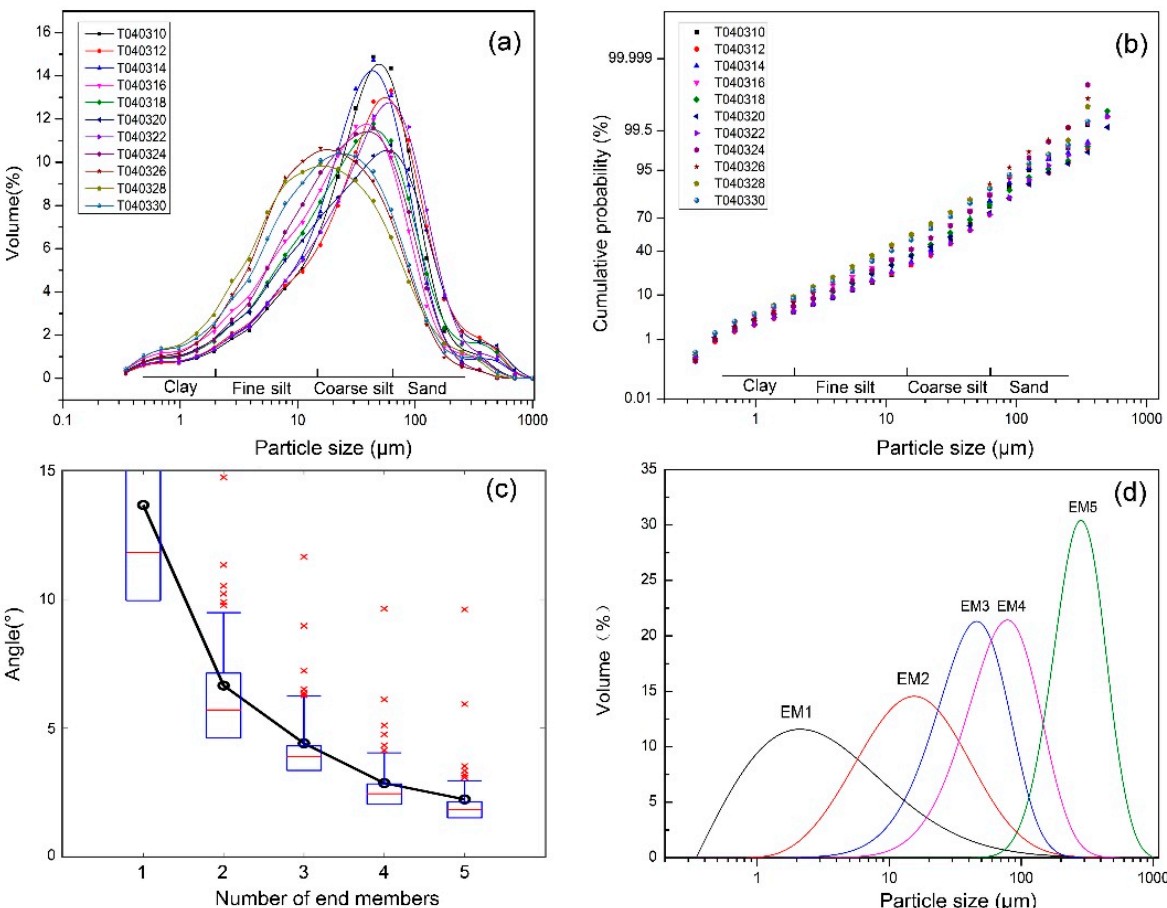

**Figure 6.** Particle-size distribution of the palaeoflood deposits in the YXP. Panel (**a**) is the frequency distribution of the particle size of palaeoflood sediments, showing that the silt fraction dominates a major role in the sediments. Panel (**b**) is the cumulative probability of the particle size of the flood layers. Panel (**c**) is the angular deviations of end members (EMs) separated from the palaeoflood deposits, the circles in the figure mean data set and the crosses for abnormal values. Panel (**d**) is the extracted five EMs, reflecting the types of major sediment components and indirectly indicating the variably hydrodynamic contexts.

## 5. Discussion

### 5.1. Environmental Context of the Palaeoflooding

During the Holocene megathermal period ca. 9.0–5.2 ka BP, during which a number of Neolithic cultures worldwide were evolving, a rapid cooling event that lasted for ca. 300 years occurred, and it was named "the 8.2-ka event" [55–57]. The curves of magnetic susceptibility and Rb/Sr ratio along the sedimentological sequence (Figure 5) both display troughs recorded at Layers 28 and 30. At around 7600 and 6700 a BP significantly climatic fluctuations were documented by the $\delta^{18}O$ variation of stalagmite from the Dongge Cave (Figure 7a [28]), and by the $\delta^{13}C$ decrease (Figure 7b [25]) in peat cellulose in the Shennongjia (Figure 7c [58]). Being an indicator of thermophilus environment, the Pinus massoniana pollen percentage (Figure 7d) is, hence, employed to reflect the palaeo-climatic suitability for dwelling at the Yuxi area. The aforesaid climatic anomaly intervals are also in response to the Pinus pollen contents that prove that at the onset of the YXC, there had always been an array of climatic anomalies comprising abrupt coolings or prolonged droughts. The adverse environmental events must have inevitably influenced the subsistence of the Yuxi Society.

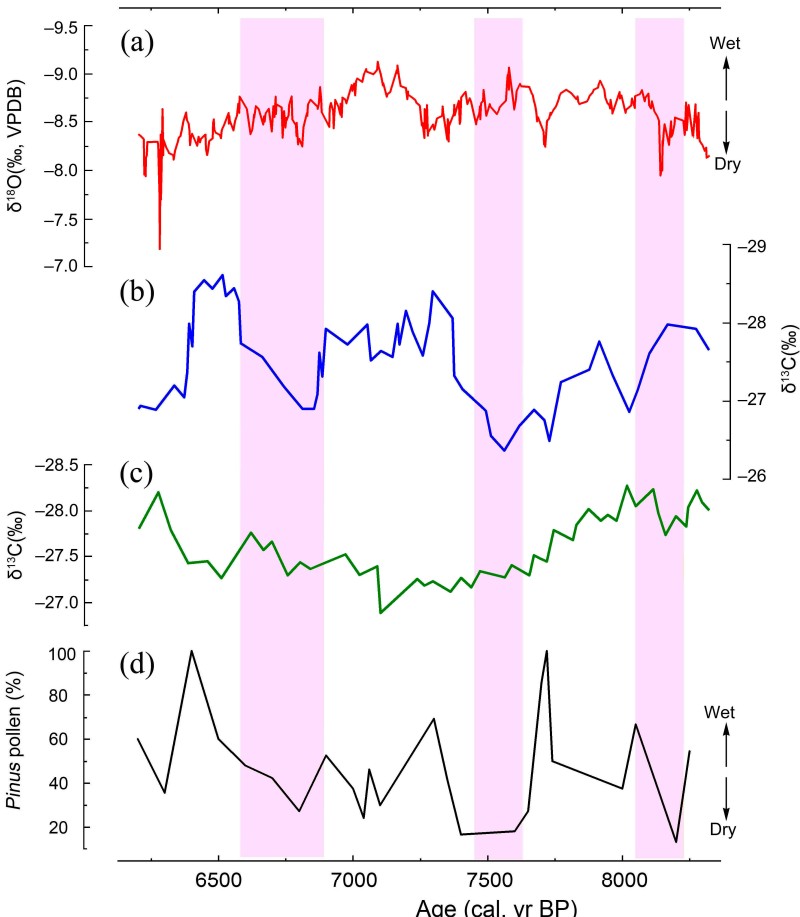

**Figure 7.** Climatic fluctuations at around 8200, 7600, and 6700 a BP recorded by different palaeoenvironmental proxies. (**a**) $\delta^{18}O$ records of stalagmite from Dongge Cave of Guizhou [28]. (**b**) $\delta^{13}C$ records of Hongyuan peat in Western Sichuan [25]. (**c**) $\delta^{13}C$ records of the Dajiuhu peat in the Shennongjia of Hubei [58]. (**d**) Pinus pollen content variation during the lower stage of the YXP. The adjacent purple bars show the marked climatic fluctuation intervals with a temporal spacing, averaging ca. 700–750 years.

The Yuxi Site sits between the Indian summer monsoon and the East Asian summer monsoon, which are closely linked to the El Niño-La Nina events both in the present and the past [25]. In case a severe El Niño occurs, the Indian summer monsoon will be weakened and the East Asian summer monsoon will be strengthened, which will lead to a spate of dreadful floodwaters [59,60]. With an enhanced El Niño, westward extension of the crest of the Pacific subtropic high occurs and induces an array of cold vortexes occupying this region [61]. Under such circumstance, the repeated occurrences of floods would be sustained for some time, probably analogous to some stages with climatic fluctuations during 8200–6400 a BP. In so doing, the 8.2-ka cooling event, well-recorded by the palaeoflood layers at the YXP, helps us understand the rules of palaeofloods in the upper Yangtze River. Even as the intervals displayed in Figure 7 (the purple bars at 6800, 7600, and 8200 a BP) roughly describe that the time span between adjacent troughs is ca. 700 years, unfortunately, since 6 ka BP onwards, the absence of flood layers in the YXP prevents us to further track the long-term periodicity for the palaeofloods.

### 5.2. Periodicity of the Palaeofloods

To see the climatically rhythmic phases that depend on the palaeoflooding durations, a particle-size sequence of the sediments of Layers 8–31 was conducted on Matlab 8.3 platform developed by MathWorks Inc. using the accessory of the wavelet tool-box. Results based on short-window Morlet Complex Wavelet (Figure 8a) show that: (i) The apparent

periods include the minor cycles, ca. 30–35 years and 6–10 years, within the high-power intervals that cover intervals 7.3–7.7 and 6.7–6.8 ka BP (the closed contours). This short flood cycle is consistent with the gauged records [39] and the short-cycled palaeofloods resulted from the long duration of La Nina events have been proven by the variation of the Dassop ice core in the Himalayas [40]; (ii) The dense contours are centered at 6.8, 7.5, and 8.3 ka, meaning that the occurrence of the palaeofloods may have restarted every ca. 750 years, as the climatic anomalies coincide with the Eddy cycle, because of a weakening of solar irradiance [62]. In addition, the power spectral density (Figure 8b) for the sedimentation rhythm shows that the peak intervals are centered on the range with frequency less than 1 Hz, where the three peak episodes can be clearly discerned at 6.8, 7.5, and 8.3 ka BP, indicating a long-term cycle for palaeoflooding under the control of the Indian summer monsoon [28] prompted by mesoscale convective anomalies and induced by La Nina events [34,63].

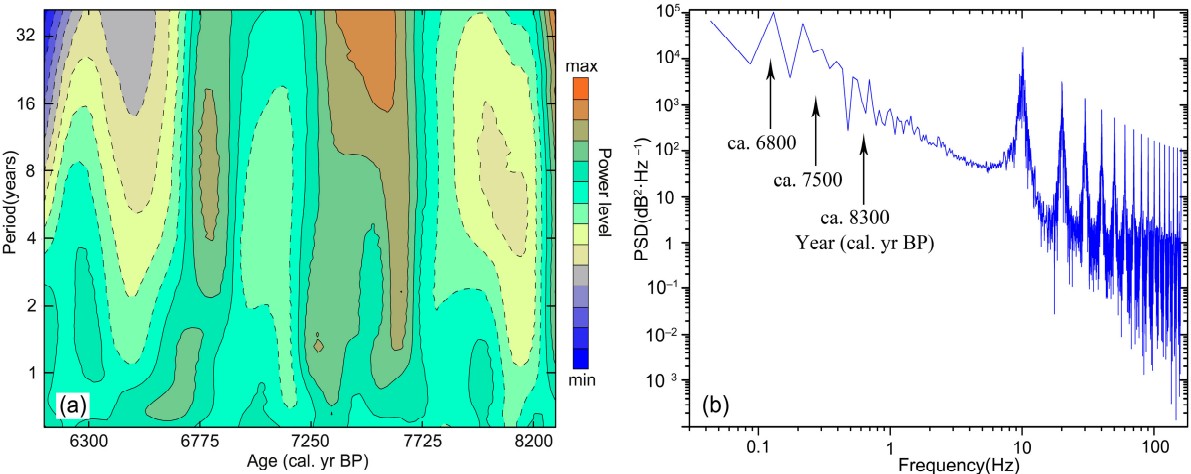

**Figure 8.** Contours of real parts of wavelet coefficient (**a**), and power spectral density (PSD) of the mean particle-size sequence for the palaeoflood sediments at the YXP (**b**), showing the periodicity of the palaeoflooding. Panel (**a**) indicates two high-energy intervals covering ca. 6700–6800 a BP and 7300–7700 a BP. Panel (**b**) indicates that the high PSD is distributed in the low-frequency interval, reaffirming the repeated palaeofloods occurred at ca. 6.8, 7.5, 8.2 ka BP. The above power is expressed as normalized decibels.

Sedimentologically, the higher accumulation rate (Table 4), except from the cultural layers, occurred during the temperate-dry. Taking the frequency of the palaeofloods into account, the maximum of which apparently coincides with the temperate-dry phase, which is evidently indicative of the Yuxi area, the frequent palaeoflood events are closely associated with the cooling events at 6.8, 7.5, and 8.3 ka BP.

**Table 4.** Accumulation rate of the palaeoflood layers during various climatic stages.

| Climatic Stage | Total Thickness of Alluvia/cm | Number of Flood Layers | Duration/Year | Accumulation Rate/cm per 100a |
|---|---|---|---|---|
| Warm-dry | 43 | 2 | 500 (6.2–6.7 ka) | 8.6 |
| Temperate-dry | 60 | 3 | 400 (6.7–7.1 ka) | 15.0 |
| Warm-wet | 99 | 6 | 1100 (7.1–8.2 ka) | 9.0 |

### 5.3. Rise of the Yuxi Culture

Although climatic change does not essentially determine the trajectory of a society, it profoundly intervenes with subsistence and social management [64,65]. The Yuxi Culture, which was mainly characterized by human artifacts, e.g., potteries, animal bone middens, burnt earth particles and charcoals, belongs to the early Neolithic type (ca. 8.0–7.0 ka BP)

coetaneous with the Chengbeixi culture, a Neolithic culture to the east of the Three Gorges of the Yangtze River. Zou and Yuan [66] argue that fishing and hunting played a dominant role for the residents as evidenced by a large number of human artifacts, e.g., stone tools, fish bones and animal bones. The identifiable animals mainly comprise Rusa unicolor, Muntiacus muntjak, Ursus thibetanus, Felispardalis, Rhinoceros unicornis, and Macaca mulatta, exhibiting a suitable habitat covered with dense shrubs and forests [67]. The Yuxi settlers also consumed a great amount of freshwater fish and mussel. Seed-floating results from Ma et al. [68] show that the crop seeds were scarcely found, with the exception of charred chips and plant seeds in the lower layers of the YXP, indicating that planting had not yet emerged. Archaeological records [69] suggest that all the found pottery shards are coarse sand-tempered, exfoliation-laminated, and made by means of mud-sheet sticking craft, reflecting that manufacturing was still in its infancy. However, the absence of palaeoflood deposits since 5.8 ka BP onwards implies that human activity intensified, exacerbating soil loss and slope-sliding.

Mainstream opinions espouse the view that abrupt climate changes resulted in the collapse of palaeo-civilizations [70–73]. Holling [74] contends that a society could successfully vanquish the external hits by forming adaptability based on panarchy of adaptive cycles. Enhancement of social systematic resilience achieved by way of tool invention and social change defines an expected sustainability [75]. Despite being influenced by the 8.2-ka cold event, this region produced to the earliest Neolithic culture in Chongqing, which was beyond the reach of the Levant and Maya [13]. The Yuxi residents had, moreover, begun to raise pigs [67] and to make tools, e.g., stone axes, stone adzes, and bone arrowheads [66,76], suggesting that hunting may have sustained the development of the community. Meanwhile, palaeofloods occurred in short cycles, ca. 30 years, or in long ones, ca. 750 years; the periodicity in this case is evidenced not only by the fluctuation of cultural succession, but also by the form and quantity of the cultural relics unearthed at the Yuxi site. As a cultural inheritance of flood-resistance, the stone-carving records in Fuling City have been warning the local residents. Apparently, the repeated palaeofloods are not a barrier but a driver for the rise of a Neolithic culture [77–80], which is obviously contrasted to the circumstance of occasional catastrophic flood events that occur in dry alpine regions [81].

## 6. Conclusions

The sedimentary features of the palaeofloods and cultural layers at ca. 8.2–6.4 ka BP in the Yuxi archaeological profile are described through the environmental proxies, i.e., particle size, element content, magnetic susceptibility and pollen composition, and the correlation between the periodic hydrothermal changes and the accumulation of the alluvia has been revealed. Additionally, a comparison has been made between the stage scenarios of cultural relics and palaeofloods. We suggest that: (1) The particle-size-based sedimentary analyses for the palaeoflood deposits can re-establish the processes of past flooding contexts well; (2) The variations in magnetic susceptibility and element ratios are in line with the climatic trend indicated by pollen assemblages; (3) The palaeoflooding cycles roughly match the recent bicentennial records, implying that the wavelet spectral method could meet the research need for sedimentological periodicity; (4) The resilience of Yuxi society when facing continual flooding seems firmly robust beyond expectation, and the abundant unearthed human artifacts from the cultural layers can prove this. In addition, the temperate dry interval with the highest accumulation rate of diluvia may be the result of soil erosion, the result of human activity and vegetation reduction. Nevertheless, the limited available archives and a deficiency in dating data have obstructed a wider view on the interaction between Man and Land, and it is difficult to observe more detailed variations in climate because of imprecise indicators. Moreover, incompletion of the found relics and an uncertainty of palaeoflood cycle mechanism, restrict the understanding of the relationship between disaster events and human activities. Even, a comparative palaeoflooding study using multi-site stratigraphic data in the area might obtain a more comprehensive conclu-

sion; exactly as we expected, some advanced methods that could help us with practical and objective insights are emerging [82].

**Author Contributions:** Conceptualization, Z.L. and W.L.; methodology, W.L.; software, W.L.; validation, Z.L. and W.L.; formal analysis, Z.L.; investigation, Z.L.; resources, Z.L.; data curation, Z.L.; writing—original draft preparation, W.L.; writing—review and editing, Z.L.; visualization, W.L.; supervision, W.L.; project administration, Z.L.; funding acquisition, Z.L. All authors have read and agreed to the published version of the manuscript.

**Funding:** This work was funded by the Natural Science Foundation of China (90411015, 41541005).

**Data Availability Statement:** Data related to this article are available to readers by contacting the corresponding author.

**Acknowledgments:** The authors appreciate the great help of Bai J. of the Chongqing Institute of Archaeology during our field investigations. Thanks must be given to Xu W., Yin Q., Guan Y., and Tian for their excellent work in field sampling and the pretreatment of the samples. We are also very grateful to the anonymous reviewers for their valuable suggestions.

**Conflicts of Interest:** The authors declare that they have no known competing financial interests or personal relationships that could have influenced the work reported in this paper.

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
