# Peer review of "Repeated Palaeofloods of 8.2–6.4 ka and Coeval Rise of Neonatal Culture in the Upper Yangtze River, China"

_sustainability, doi:10.3390/su15010187_

Round 1

Reviewer 1 Report

The paper entitled ‘Repeated palaeofloods of 8.2–6.4 ka and coeval rise of neonatal culture in the upper Yangtze River, China’ the authors analyzed a complete sedimentary sequence of alluvia found in Yuxi profile, the paper was well written and easily accessible.

1. Figure1. The geographical location of Yuxi, the legend of Figure 1 is incomplete, the north arrow and the scale of the map should be added.

2. could the authors list the distribution of the soil samples on the map? Or provide more information about the soil samples? Soil collection standards or different categories?

Author Response

We have made revisions to the manuscript according to your guidance and suggestions. The detais is encolsed in the attachment.

Reviewer 2 Report

1.      I suggest the author demonstrate what does the paper add to the current literature? and what new knowledge is added by this study?

2.       It is suggested to present the structure of the article at the end of the introduction. At the end of the introduction add a para including 1-Gaps in the backgrounds you try to compensate them, 2-your novelty and unique aspects 3-Hypothesis 4-Objectives.

3.       Discuss the merits and limitations of the technique applied.

4.      The presentation fails to discuss the summary and tries to some of the vague reasons, which is not the explanation. Need to compare the results with new references.

5.       The explanation for the critical analysis is not sufficient, although some of the good points have been identified.

6.       Please rewrite the conclusion with the proper explanation in the R & D and innovation process.

7.      Abbreviations are numerous in this manuscript! They should be explained before the introduction.

8.      The material and method section is too weak in the manuscript and you need to focus on it more.

9.       Please revise your conclusion part into more detail. You should enhance your contributions, hypothesis retain/reject, limitations, implications/applications, advantages/disadvantages, policies, underscore the scientific value added to your paper, and/or the applicability of your findings/results and future study in this session

Author Response

We have completed revisions of the manuscript.

Reviewer 3 Report

The manuscript with this title repeated palaeofloods of 8.2-6.4 ka and coeval rise of neonatal culture in the upper Yangtze river, China and result is interesting but the manuscript needs to be improved.

The Introduction should be writing topic -based, and I recommend improving the literature and comprehensive introduction.

 Figure1. The map ‘s quality and some detail should be improved. Lat and Lon 

It is necessary to include information about the plant cover in the study area as well as pollen.

The order of methods should be explained better; I think it would be better to show them in a flowchart.

A soil profile illustrating the sequence of soil.

A flood history in the study area.

Author Response

We have made a major revision of the manuscript.

Round 2

Reviewer 3 Report

Thanks for revising the manuscript. As a minor revision: 

 The Fig 5 quality  is rather weak. An english grammar and spelling check is required.

Author Response

Please see the attachment below.
